# Time-Dependent Development of Scientific Discourse: A Novel Approach Using UMAP and Word Embeddings

Jonah Lynch

September 18, 2024

### Abstract

This study presents a novel method for visualizing the time-dependent development of a scientific discipline using UMAP (Uniform Manifold Approximation and Projection) and word embeddings. By encoding the abstracts of scholarly articles into a high-dimensional space and then projecting them into a 3D space, we demonstrate how the evolution of research interests and topics in a specific field can be mapped over time. This approach offers new insights into the dynamics of scholarly discourse and the emergence and disappearance of research themes.

## 1 Introduction

Scientific publications present a constantly evolving profile, with new topics emerging and others fading from prominence over time. It is difficult to capture the complex, multidimensional nature of scholarly discourse because of its sheer quantity, and any attempt to describe and evaluate it must struggle with weighty interpretative issues. Each writer has a point of view, a specific cultural context, and prior commitments that may not even be conscious, but which heavily influence the overall patterns they claim to discern in the development of a discipline.

Many scholars have paid close attention to this issue. One, a digital humanist named Johanna Drucker, has carried out several projects that aim to interpret data without reference to external sources of truth. These projects help situate the proposal made in this paper. An approach which Drucker called temporal modeling attempted to take into account the multiple values time has for different observers without assuming that time is a common reality to all observers. She explains that "Our goal (never quite fully achieved) was to construct a system that was not based on a container model (time as a preexisting framework into which events or incidents were put) but based on relationality. The proportions and scales of the temporal model would emerge as events and references were put into relation with each other."[2] In other words, instead of positing an external point of reference such as a "second" or an "hour", she meant to construct the concept of time entirely through internal references between texts. Drucker was certainly correct in challenging the idea of time as a "container", which in the natural sciences is a concept that has been superseded by more nuanced depictions. But she says this goal was never fully achieved. The reason might have to do with the fact that in order to compare two quantities, n the absence of a direct, physical comparison between two objects, an external metric is usually necessary. An external measure, such as inches or centimeters, does permit comparison.[1] To be sure, it does do so not without imposing strictures of its own, such as the tendency to express things in multiples of 5 or 10. It might also impose value judgments about the nature of space, which might need to be specified, and perhaps there would also be an anticolonialist critique lurking somewhere, along the lines of Drucker's point about the Celsius scale not taking into account the experience of cultures that had never encountered freezing temperatures until recently.[2] But it might be the best

---

[1] In physical space, time is intimately linked to gravity and acceleration in general; in biological space, an animal lifetime can be closely correlated to the total number of heart-beats. The psychological human experience of time is an even more complex issue.

[2] The Centigrade temperature scale is constructed on the basis of an experience (boiling and freezing water) which some peoples had never had before refrigeration and travel made it available to them. This example is susceptible to further critique, since the experience described is also as "objective" an event as can be found. Emphasizing the cultural contextuality of this measurement scale might be pushing it a bit. Water does not freeze or boil at arbitrary or culturally-determined temperatures.

one can do. Drucker emphasizes what she calls a "nonrepresentational approach". "A nonrepresentational approach does not preclude the use of an image to depict or figure something; it simply suggests that an image cannot represent something in a relation of equivalence." Of course, "it is a truism that a map is not equivalent to a territory." But consider a space as inhabited by a "small child, a threatened woman, a preoccupied man" and the meaning of the details of the space they inhabit change radically. That meaning is not represented within the map: it must be reached through interpretation. "Interpretation is a constitutive act, one that makes the object of investigation through engagement," she claims. Therefore, "the challenge is to understand how to model interpretation computationally."[2] Drucker conjures the Vannevar Bushian image of a 'memex' that can "integrate the activities of scholarly research—bibliographical trails and links, text analysis, visualization, multiple (many) documents, and the layering of interpretative frameworks onto the underlying evidence." It is true that even the most extensive collection of data points is only a small subset of the information that could be discovered in the future. The data we currently possess—all the patterns we have—are like a few branches illuminated by a campfire, while out beyond the flickering light of our intelligence lurk the unexplored forests and trackless prairies of the Being we have not yet encountered. In the semi-darkness, we tell stories and make models. The aim, as G. Buccellati put it in his *Critique of Archaeological Reason*[3], is to produce "a cognitive structural whole that is self-contained" and which can be "understood and described on its own terms, without reference to anything external to it." This article describes a novel method to produce such an "inner-referential", self-contained structural whole in order to model the development of scientific discourse.

# 2 Methodology

## 2.1 Data Collection

Several large databases of open-access scientific articles exist. In some preliminary experiments leading up to this article, the JSTOR repository was used in conjunction with the Constellate service, which allows users to filter articles and download a json file containing metadata, and in some case, full article text. This choice was initially made in order to study historiography and other humanistic disciplines using digital methods. However, temporal coverage in the JSTOR archive is variable, and some journals and subjects are present with more articles toward the beginning of the 1900s than later on, which clearly does not correspond to the monotonically increasing quantity of scientific papers published over the course of the last hundred years. Some reasons for this are the ways the JSTOR archive has been compiled, and the fact that access is partially metered according to the credentials of the user and their hosting institution. Furthermore, although the articles and their metadata are presented in a json, many of them appear to have been extracted using older OCR techniques and contain many errors in the text. It proved difficult to extract a file that could be reasonably expected to proportionally represent the development of a field for the purposes of this article.

Another archive of open-access scientific articles which has better characteristics for this study is hosted at Cornell University: the arXiv. This repository began in 1991, and originally arose as a way to share physics research much more quickly than the traditional publishing system allows. Today it also hosts articles on Mathematics, Computer Science, Quantitative Biology, Quantitative Finance, Statistics, Electrical Engineering and Systems Science, and Economics. Of these, the first three disciplines are by far the most heavily represented.[3] The arXiv can be queried through an api, and it also permits bulk download of a large quantity of articles which can then filtered locally.[4]

After several experiments to determine the most useful subset of this data, I extracted 55,667 articles tagged as Computer Science - Computation and Language (cs.CL), dating from 1995 to 2024. This domain has several characteristics that make it amenable to study as a time-dependent phenomenon. Natural language processing has been an important and highly studied issue in computing since before the 1990s, so there is a long-term signal that can be seen in the development of publications on the topic. NLP is also a subject that has known periods of "boom" and "bust", and at at any given time, only a subset of the many techniques that have been tried remain in use. There are also some seminal moments in the field where one might expect a new period of "boom" to begin, such as the 2013 publication of *Word2vec*.[6] The hypothesis,

---

[3]See https://info.arxiv.org/help/stats/2021_by_area/index.html
[4]The easiest way to download this information is at https://www.kaggle.com/datasets/Cornell-University/arxiv

therefore, is that a spatial representation of the information contained in article abstracts in the domain of Computation and Language will show, to some degree of precision, features that can be correlated with events in the field.

## 2.2 Text Embedding

In order to represent articles in a space, several decisions had to be taken. First, which text should be used as a proxy for article content? Would the title suffice, the abstract, or should the full text be used? While this may not be true of all sciences, in Computer Science, article abstracts often provide a trustworthy and highly compressed representation of the article contents. For computational efficiency and the epistemological need to focus the representation of each article into a single vector, I chose in this case to forego examination of the full text of the articles. It is clear that this choice carries important consequences, and should be revisited in any future application of this method. Here, it seems a reasonable first-order approximation in order to introduce a new method.

Each abstract was embedded using the avsolatorio/GIST-Embedding-v0 model[7], a near-SOTA quality open-source model that particularly excels at semantic textual similarity tasks. New embedding models are released continually, and the aim of this study is not to compare them. In the future, it would be useful to do so in order to determine which models are most useful for the kind of representation presented here. The embeddings produced by this model have 768 dimensions.

## 2.3 Dimensional Reduction

Such high dimensional vectors cannot be intuitively understood by humans, so it is necessary to apply a dimension reduction procedure in order to create a visualization. In this case, UMAP[5] was used because it is computationally efficient and represents global structure more faithfully than than two other main techniques for dimensional reduction, t-SNE and PCA. This choice depends on the consideration that distinguishing separate topics, and clustering similar topics, will be more useful for mapping their time-dependent development than approaches that capture more of the specificity of each article. Of course, this choice is open to future criticism and modification.[5] For the present purposes, UMAP projections calculated with the default parameters seem sufficient. In the first versions of this project, the vector embedding of each abstract was projected onto a 2D space using UMAP, and the time dimension (extracted from the article metadata) was represented as a color gradient. However, even a small number of articles produced messy visualizations that, while visually pleasing, were difficult to interpret. A second visualization attempt represented the time dependent character of the embeddings as a video. While suggestive, this approach was also too difficult to interpret, in part because there was no convenient way to explore the specific contents of the clusters of points projected into the space. The approach presented here uses 3D plotting to represent the structure of UMAP reduced embeddings in the X,Y plane, and time on the Z axis. The resulting volume can be easily viewed interactively on a computer screen.[6]

## 3 Results

Figure 1 shows the results of the above method. The projection to 2D using UMAP defines the X,Y coordinates of each point, and time is represented on the vertical axis. A few features are immediately evident from this visualization. At the beginning of the dataset, in the year 1995, there was more publication in the cs.CL domain than later on, around the year 2005. Later, between 2010 and 2015, a large increase in the number of publications occurred. If we rotate, zoom, and examine more closely, other features are visible.

At the left of figure 2, five dots corresponding to five articles can be observed. All five articles have very similar embeddings in the 2D space, and are therefore plotted in a vertical 'column'. By mousing over, the viewer can note that all five articles deal with explanation-based learning, a technique used in NLP in the 1990s, but which has not been developed further in more recent years.

---

[5]See for instance the helpful interactive visualization of a mammoth in 3D and in 2D UMAP projection at https://pair-code.github.io/understanding-umap/

[6]A sample visualization can be found at ████████████████████████████████████████████████

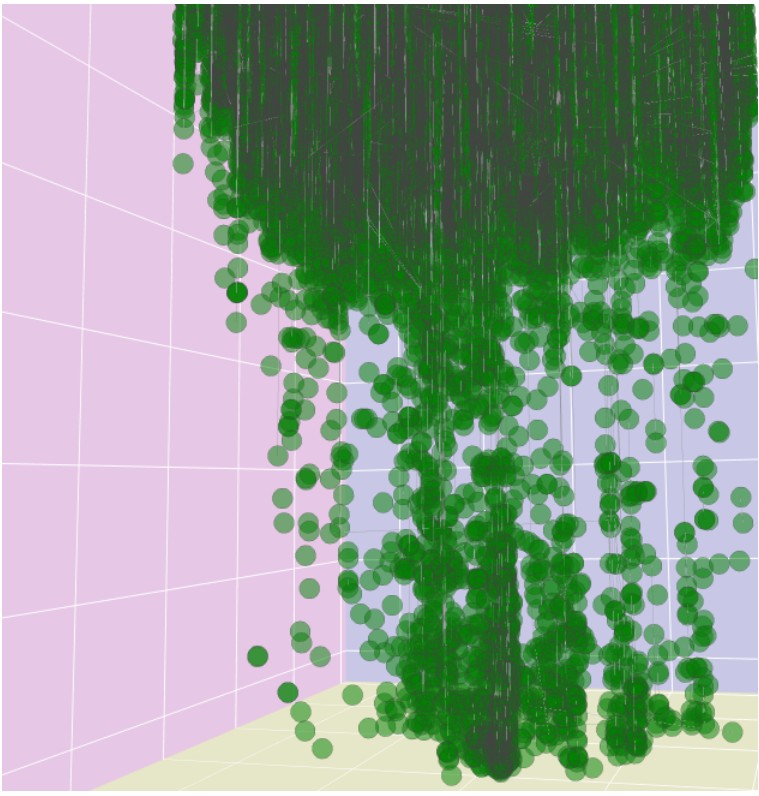

Figure 1: UMAP projection of abstracts of cs.CL articles, with time on the vertical axis

These facts validate the initial hypothesis: without referencing any data outside of the 55k articles studied, and in particular without accessing any meta-science or historiographical research about the development of NLP, computation and visualization clearly shows that several specific field of studys were developed until a certain time and then no more. On a longer time scale, the same representation also shows that something happened around 2013 which led to an explosion of interest in the field after a period in which publications were much more scarse. Was it *Word2vec*? Correlation is not causation, but intuitive access to the fact that something changed around that time can be helpful for a scholar wishing to study the temporal development of the field. By further examining the structure of the visualization, an interesting feature can be found in 2014: a very dense 'column' structure which did not previously exist begins with an article entitled *Deep Fragment Embeddings for Bidirectional Image Sentence Mapping*[4]. The use of RNNs to process images and text description too is a domain of research with explosive growth over the last decade. Here again the topology of the representation allowed identification of an interesting feature that began at a a certain date, in a certain domain. Of course, manual exploration of the graph is difficult when there are a large number of articles present. Further methods of analysis of this representation are a subject of continuing research.

## 4    Discussion

This study began with the idea of "inner-referentiality", the attempt to construct models of the world without reference to external metrics or sources of truth. Much research has supposed that by encoding data about the world into symbols that can be manipulated by a machine, a representation of the world is created within the machine itself. This approach may run aground, however, upon the circular logic it seems to imply: "encodings can only transform, can only encode or recode, representations that already exist." [1]. According to Mark Bickhard, in order for an encoding to actually represent an element of the world, a meta-representation is required. Such an affirmation seems to imply a logically inconsistent infinite regress or an "unmoved mover" that is not itself inside the system. Many researchers have assumed that

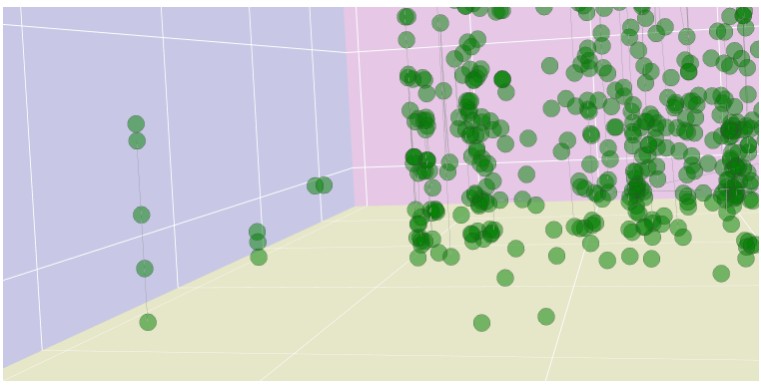

Figure 2: Close-up of the early articles in the dataset, showing a 'column' feature

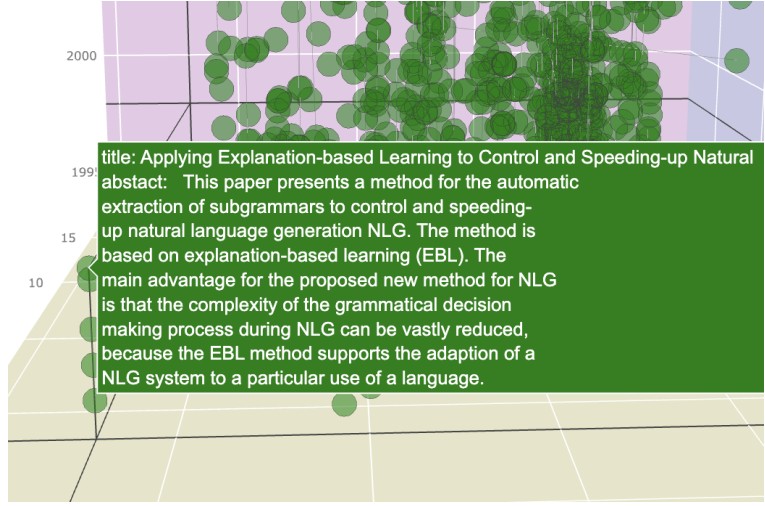

Figure 3: User interaction with the zoomed image

the problem is not as serious as this, and expect a large quantity of information to converge on a machine representation of experience from which further experiences can be predicted. In the domain of robotics, Bickhard's criticism might hold, since "so long as our modeling vocabulary is restricted to such factual correspondences, there is no way to provide (to an agent) knowledge of what the correspondences are with. It is crucial to realize that knowing that something is in correspondence and knowing what it corresponds to is precisely one version of the general problem of representation we are trying to solve! Thus, as an attempt at explaining representation, encodingism presupposes what it purports to explain."[1] But in a research tool such as is presented here, Bickhard's criticism does not hold. Here, words are translated into mathematical representations, high dimensional vectors which can be thought of as "encoding" some degree of the semantic and syntactic content of the original text. Because they are mathematical objects, they can be robustly compared with a variety of tools. The act of comparison and analysis can lead a scholar to insights that would otherwise be difficult to achieve, because it is difficult for a humanh to read and remember the contents of 55 thousand articles. Are these comparisons "inner-referential", because there is no apparent ground of truth outside the system to which the words or the vectors can be compared? In fact, there is an external factor which must be considered, and which is a crucial feature: the language model which is used to generate the word vectors in the first place. From an information theoretical perspective, language models compress the information content of a large amount of text. Therefore, the word vectors generated by embedding article abstracts within the space of a language model contain information about the embedded text as it exists in relation to all the text that was used to create the language model. In themselves, word vectors are not inner-referential, because they are defined through comparison with all the

text that generated the model. The overall system *is* inner-referential however, in the sense that the entire body of text used to train the model, plus the articles embedded within it, are not defined in reference to an external ground truth.

## 5   Conclusion

The method presented in this study offers a novel way to visualize and analyze the development of scientific discourse over time. This method combinines word embeddings and UMAP dimensionality reduction in a tool that can reveal patterns and trends in scholarly discourse that might otherwise remain hidden. This approach has potential applications not only in meta-science and historiography but also in any field where the evolution of ideas is of interest. Source code for producing the visualizations shown here can be found at ▨▨▨▨▨▨▨▨▨▨▨▨▨▨▨▨▨▨▨▨▨▨▨▨▨▨▨▨▨▨▨▨▨▨▨▨▨.

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
