# OpenReview forum: "Time-dependent Development of Scientific Discourse: A Novel Approach Using UMAP and Word Embeddings"
_ICLR.cc/2025/Conference — Submitted to ICLR 2025_

### Official Review · Reviewer_Zu11 · 2024-10-15

**Soundness:** 1
**Presentation:** 2
**Contribution:** 1
**Rating:** 1
**Confidence:** 4

**Summary:**

The paper proposes a method to visualize the evolution of scientific fields over time using UMAP (Uniform Manifold Approximation and Projection) and word embeddings. The authors extract data from scholarly abstracts and project them into a 3D space, representing topics' temporal development.

**Strengths:**

* The use of word embeddings and UMAP is valid, especially for large-scale textual data visualization.
* The approach could potentially serve as a useful tool for meta-scientific analysis or historiography.

**Weaknesses:**

* The choice of embedding model is not adequately justified, nor are comparisons made with other models to strengthen the findings.
* The visualizations lack clarity, making it difficult to extract meaningful information from them.
* The paper does not engage deeply with related works, missing out on critical discussions about existing methods for analyzing research trends.
* The results fail to provide actionable insights and the paper misses contributions.

**Questions:**

- Could you provide more justification for using the selected embedding model? How do you know it is the most appropriate for this task?
- How do the visualizations concretely support your claims about the evolution of research topics? Could you provide clearer examples?
- Have you considered comparing the results of UMAP with alternative dimensionality reduction techniques such as PCA or t-SNE?

**Details Of Ethics Concerns:**

no double-blind submission

---

### Official Review · Reviewer_gPJ3 · 2024-11-04

**Soundness:** 1
**Presentation:** 1
**Contribution:** 1
**Rating:** 1
**Confidence:** 5

**Summary:**

This paper presents an exploratory analysis of the embedding of paper abstracts over time. The author embedded the abstracts of 55k articles from arXiv (cs.CL), between 1995 and 2024, using GISTEmbed model. The embeddings are then projected onto a low-dimensional space with UMAP, with another axis representing the time. Qualitative exploratory analysis is performed.

**Strengths:**

The text embedding-based approach to study scientific discourse and evolution is a promising direction.

**Weaknesses:**

Unfortunately, I believe that the paper is not a good fit for the ICLR conference. The paper does not make a significant contribution to the field of representation learning or machine learning. The paper presents a qualitative exploratory analysis of the embeddings of paper abstracts over time, which does not represent a significant advancement from the common practice of analyzing embeddings that are performed routinely in various contexts.

For instance, conferences like NeurIPS have been providing embedding-based visualization tools for multiple years. Companies like nomic.ai have been providing the temporal analysis of scientific literature over time at the scale of millions of papers (https://www.biorxiv.org/content/10.1101/2023.04.10.536208v1). Research groups like AllenAI, Northwestern University (e.g., Dashun Wang), University of Chicago (James Evans), Indiana University (YY Ahn), Northeastern University (Albert-László Barabási), Stanford University (Dan Jurafsky), and many others have been extensively employing paper embeddings and other related methods to study scientific discourse and evolution. Given this long list of existing work (with much more sophisticated analyses) that I can't list all here (and not cited by the paper), the manuscript would need to provide a lot more results and analysis to make meaningful contribution. I'm sorry to say that the paper misses the mark quite substantially in this regard.

An important issue that the author should also pay attention to is that we shouldn't simply use the UMAP axes as meaningful coordinates. This has been extensively critiqued by Lior Pachter and other researchers (e.g., https://www.biorxiv.org/content/10.1101/2021.08.25.457696v1).

**Questions:**

The paper would need a thorough survey of existing literature. The current version misses pretty much all important literature in the field of science of science and scientometrics, particularly the extensive usage of embedding in these fields.

I think the paper would need a clearer research question and a concrete, quantitative analysis.

However, even if these issues are addressed, I think the paper would not be a good fit for ICLR.

---

### Official Review · Reviewer_N58k · 2024-11-04

**Soundness:** 1
**Presentation:** 1
**Contribution:** 1
**Rating:** 1
**Confidence:** 5

**Summary:**

As per my previous comment, this paper is not anonymous and should be desk-rejected. Thus, I did not perform a full review. If there should be one required anyways, please send a respective comment.

**Strengths:**

-

**Weaknesses:**

-

**Questions:**

-

**Details Of Ethics Concerns:**

No anonymous submission.

---

### Meta-Review · Area_Chair_1XrW · 2024-12-16

**Metareview:**

All the reviewers agree to reject the paper due to its very limited novelty and technique contribution. The paper is not anonymous and should be rejected.

**Additional Comments On Reviewer Discussion:**

The authors did not provide rebuttal, and thus there is no discussion.

---

### Decision · Program_Chairs · 2025-01-22

Reject